# The Concept of a Substance and Its Linguistic Embodiment

## Henry Laycock

Emeritus (Retired), Department of Philosophy, Queen's University, Kingston, ON K7L 3N6, Canada; laycock-queens@proton.me

**Abstract:** My objective is a better comprehension of two theoretically fundamental concepts. One, the concept of a *substance* in an ordinary (*non*-Aristotelian) sense, ranging over such things as salt, carbon, copper, iron, water, and methane—kinds of stuff that now count as (chemical) elements and compounds. The other I will call the *object-concept* in the abstract sense of Russell, Wittgenstein, and Frege in their logico-semantical enquiries. The *material* object-concept constitutes the heart of our received logico/ontic system, still massively influenced by Aristotle after almost 2.5 millennia. On such an account, the fundamentality of material objects and their attributes are the metaphysical basis of the cosmos, as reflected in our received logic, Quine's 'canonical notation'—derived via the empiricism of Russell from Frege's function-based *Begriffschrifft*, and consisting of concrete singular terms and variables, quantifiers and predicate-expressions. The inadequacy of Frege's approach to understanding concepts is reflected in his initial question. Frege enquires of 'what it is that we are calling an object', remarking that he regards a regular definition as impossible: "we have here something too simple to admit of logical analysis". The imagined ultimacy or simplicity of the idea of a *single* object (arithmetically, just *a unit*—one as opposed to two, three, four, etc.) as foundational to the calculus is just that—imagined. It is also guaranteed to block the comprehension of the substance-concept.

**Keywords:** concept; substance; Frege; ontology; object; Aristotle; Quine; Presocratics; abstraction; category

---

'Rules and formulas, those mechanical aids to the rational use, or rather misuse, of one's natural gifts, are the shackles of a permanent immaturity'

Immanuel Kant, *What is Enlightenment?*

'The power of reason must be sought not in rules that reason dictates to our imagination, but in the ability to free ourselves from any kind of rules to which we have been conditioned through experience and tradition'

Hans Reichenbach, *The Rise of Scientific Philosophy*

## 1. Introduction: Some Historical Observations

The scheme of categories famously outlined in Aristotle's *Categories* is grounded on a single basic contrast, that between concrete individual *things* or 'primary substances' (individual men, individual horses), and a small range of general categories, comprising both *concrete-substantive* related 'secondary substances' or *kinds* (Man, the horse), and *concrete-adjective* related *attributes* (solidity, hardness). Alongside these general categories, however, two other, more narrowly focused categories are subsequently introduced via the *Metaphysics*, apparently to better comprehend the primary substances themselves—the contentious pair of 'hylomorphic' categories—for a bronze sphere, the categories of *matter* (*hyle*, wood, bronze) and *form* (*morphe*, structure, sphere).

Notwithstanding the remarkable growth of knowledge between the classical epoch and today's modernity, Aristotle's scheme has demonstrated an extraordinary resilience or durability—and, evidently, too, a substantial and persistent range of followers.[1] At the same time, it would be surprising indeed if the considerable growth of knowledge across these epochs—accompanied as it has been by an overall deepening and transformation of our

taxonomies of nature—did not call for ongoing readjustments to the Aristotelian scheme itself. Indeed, just such appears to be the case with the category of *matter* in particular—the understanding of which has manifestly deepened significantly over modernity, in particular since the empirical and quasi-taxonomic work of Lavoisier and Mendelayev. It is by no means inconceivable that such an understanding could be sufficient to undermine the supposed primacy of 'primary substances' themselves.

Hence, among the central notions here in question is precisely the everyday conception of a 'substance', as applied to a theoretically robust range of kinds of *stuff*—such things as carbon, silver, iron, salt, and water. It goes, I think, without saying that the notion itself is neither a notion derived from chemistry in particular, nor from science in general. Rather, it is a notion inevitably present in everyday thought and talk from times immemorial—talk of *water* in particular must surely be as old as *Sapiens* themselves. The concept can I think be recast in terms of the notion of a range of potentially *pure* substances—ones with a robust definition and an intrinsic, qualitative homogeneity—such that the very recognition of impurities signals acknowledgement of this concept of a substance. And not only is this concept truly ancient, is also, so it seems to me, conceptually fundamental to our overall scheme of concepts (as here presented in the sequel).

But although the category before us here is 'pre-scientific'—existing and recognised as such long, long before the 'Scientific Revolution'—our understanding both of it and of its role within the cosmos have deepened enormously with the growth of knowledge. The concept first enters philosophical scrutiny hand in hand with the appearance of philosophy itself—the two are in effect coeval, from the epoch of the Ionians, earliest among the Presocratics. And the concept of a category peculiar to silver, salt, water, and the like, much as now, is quite simply this truly ancient category of 'the elements'.

The principal question of this work is then the nature of the concept of a substance, in this robust yet everyday, non-technical sense of the term.[2] And while there is no denying that we know more today about the constitution of reality than did the ancient Greeks, the ancient *pre*-Aristotelian doctrine regarding the 'elements' has turned out to be at least 25% correct. The elements and compounds of modern chemistry are the basic chemical constituents of reality. They are, as it happens, the 'underlying stuff' of which everyday material objects are composed, and they do, of course, include water—one of the four ancient elements. The reality is, however, that these substances have been cut loose from their Ionian origins. And thanks largely to Aristotle and to atomism in one form or another, their profound philosophical or metaphysical import has been lost sight of. In this work, it is my hope that we may begin to restore what has been lost.

Now there is—and long has been—a prevailing ontology or metaphysics, a doctrine I have elsewhere called 'the ontology of objects'.[3] This is the seemingly 'common-sensical' thought reflected in Quine's remark that 'we are prone to talk and think of objects'; it is also the central topic for critical scrutiny within this work.[4] The doctrine may be understood logically or semantically—as a form of talk in which concrete singular *reference* is seen to be the primary mode, not simply of reference as such, but also of communication in general, before all else can be expressed or said. But again, as a constituent of these massive historical transformations, there is the transformation of conservative logic itself—from the syllogistic subject-predicate logic of Aristotle to the algebraic/functional analysis of Frege (in numerous publications). At the same time, however, the doctrine may be understood in a more directly metaphysical manner, as a claim to the effect that concrete or material objects somehow constitute the ontological 'foundations' of the cosmos, in terms of which all other goings-on might perhaps be understood.

The doctrine is confronted, I believe, with virtually insuperable challenges; and to grasp the profound difficulties of the ontology of objects is thereby to comprehend the notion of a substance. And yet, from this latter standpoint, the crucial point is not so much metaphysical or ontic, as purely and simply *semantic*. This statement might well appear bizarre—an issue that needs itself to be both acknowledged and addressed. Being not only central to the present project but also having, in my view, surprisingly broad implications

for the understanding of related concepts such as that of causality, the claim is one that calls for validation, and which must consequently follow in the sequel.

## 2. Words and Concepts

Crucial to this project—not only generally but also in the context of our relatively new approach to logic—is the difference between words, along with their semantics, and the exploration of concepts. I would like to suggest that an adequate grasp of this distinction results in according zero special status to the notion of 'the one'—in effect, the 'singular' notion of a *single unit*. And although the object-concept is distinct from its singular and plural embodiments or expressions, there might perhaps be a tendency to conflate metaphysical or ontic and conceptual matters—a tendency to conflate the singular *expression* of the object-concept with the concept in itself.[5] And it is perhaps this that results in the supposition that not only is 'the unit' that out of which two, three, four and five number-units can be constructed, but also that the notion of plurality or multiplicity is itself derivative from the notion of a single unit. There is, I think, a strong tendency to 'ontologise' the linguistic contrast between singular and plural uses of a noun—to see the plural as a sort of extension or simple *multiplication* of what is denoted by the singular. The one, after all, is less complex, or perhaps simpler, than the many. However, the contrast of singular and plural is primarily a semantic contrast, and what it represents conceptually are distinct modes of linguistic embodiment of one and the same more abstract concept. Just how the different uses or embodiments of the concept are to be conceived remains very much an open question—thus understood, the character of the concept can only be inferred from the complete variety of its diverse modes of embodiment within a particular linguistic lexicon.[6]

Nevertheless, given the contrast of words and concepts, and the conceptual unity underlying the diversity of its numerical expressions, the number-concept must *itself* be counted as numerically neutral. Furthermore, the seductive thought that singularity is primary in some intuitive sense—*less complex* than its plural counterpart, talk of one thing 'surely' being basic or just simpler over talk of many, no doubt works to reinforce this tendency. This, however, is a strategic mistake: there is a defect in the Russell and (especially) the Frege treatments to be shortly introduced. The defect is an outcome of this very simple fact: that expressions of the object-concept that are *canonical* for these accounts—the *examples* of the object-terminology itself—are just the ones whose form is singular *exclusively*.

The issue here concerns the contrast between the semantic properties of words and the attributes of underlying concepts. Concepts in general—formal or otherwise, and the concepts of categories in particular—are quite fundamentally distinct from words and sentences. One and the same concept can be expressed in distinct forms of words—not only in different languages, but also within one and the same language—commonly enough, *both* as singular and as general terms. We are not, for example, inclined to think that 'happy', as an adjective, and 'happiness', as a noun, express different *concepts*—although the terms themselves have quite distinct semantic *functions*. Concepts, unlike words, elude direct inspection: They are concealed or veiled by words, lying behind the modes through which they are linguistically represented or expressed. Words are conventional devices; concepts and categories, so I think, are not. I shall speak of concepts, along with the categories to which they correspond and which indeed they reflect, as having semantical *embodiments* or modes of embodiment in various forms of sentences, or as being represented or expressed in the use of such sentences.[7] Concepts cannot, plainly, be represented *other than* in their modes of embodiment—they can be expressed only via the use of words or perceptible devices of some other kind. Nevertheless, it is the former, rather than the latter, that are here our fundamental objects of enquiry. Much as with 'happy' and 'happiness', there are two basic modes of embodiment of the substance-concept (singular, as with names, and general, as most typically with substance-words). And this same point is equally applicable to more general, 'category-representing' terms such as 'object', 'unit', 'entity', and so forth

in themselves. Whether we speak of *objects* or *an object*, of *units* or *a unit*, one and the same underlying category or concept is evidently expressed. This, in effect, is the force of Russell's remark in the *Principles of Mathematics* [6] when, in responding to the Leibnizian maxim that 'Whatever is, is one', Russell retorts, 'Whatever are, are many' [6] (p. 132). Objects are the ontic correlates of thing-words or count nouns, and rather as objects may be either one or many, words for objects may be either singular or plural—singular and plural are the twin semantic groups that together virtually exhaust this enormous category of nouns.

But then, whereas singular and plural are semantically (and often, too, syntactically) distinct, the corresponding ontic category or concept in this context, that of *things* or *objects* in itself, is *neither* singular nor plural—neither 'one' nor 'many', so to say—although it may be represented or expressed semantically, and perhaps syntactically, in *either* singular or plural form. And since the category as such is neither singular nor plural, it may be said to be numerically *neutral*. The object-concept might perhaps be said to be the concept of a single unit; and insofar as objects do, in fact, exist, there cannot fail to be at least a single *one*.[8] It might then perhaps be thought, on precisely this account, that the concept *cannot* plausibly be said to be numerically neutral. But the objection begs the question; for while objecthood and unity are not distinct, the fact is that exactly as *an* object is *a* unit, so *objects*, equally, are *units*; and units may be *either* one or many. Hence once again, the concept in itself appears to be numerically neutral: one and the same category or concept appears *both* in singular and in non-singular semantic forms.

A parallel judgment can, of course, be made concerning mass-words. Now the substance-concept is not only very ancient; it is also, I believe, conceptually fundamental. Being central to the present project and having, in consequence, broad implications for the understanding or related concepts such as that of causality, the claim is one I undertake to validate in the sequel. And perhaps the most obvious and relevant quasi-metaphysical fact about *water*, to take what is surely the most obvious example, is precisely that it, this very same compound, is *ubiquitous*—it is not present only in the here and now, but in some sense, is widely 'scattered' or distributed—a point remarked if not so much explored by Quine.[9] *Qua* liquid, it covers most of the surface of our planet, in the form of seas, oceans, lakes, rivers, pools, puddles, and so on. As a gas, this same substance is now known to be present not only in the atmosphere of the Earth, but also in that of the Sun, and indeed in the atmosphere of Titan. As a solid, it is present on our moon, on Mars, in comets, and so on. It extends throughout the Kuiper belt as solid chunks and grains, into the very farthest reaches of this solar system, and beyond, into inter-stellar space. It is, of course, a compound of two common elements, and the existence of this particular substance in one or more of its physically distinct states, solid, liquid and vapour, is a matter of everyday experience (there is no such experience of plasma).[10]

The idea of a scheme of metaphysical categories or high-level and perhaps *a priori* concepts, concepts such as those of object and identity, space and time, change and the causal connection between events—roughly, semantically encoded but high-level, abstract and maybe non-empirical concepts—is not unfamiliar. Arguably, it is also a scheme within which much of physical theory, and classical Newtonian mechanics in particular, is written or expressed. Thus conceived, metaphysics typically aspires to being a systematic account of what are taken to be the most basic, highly general features of reality itself. Then, as the core theory of 'being' pure and simple—in effect, the theory of 'what there is'—the narrower domain of ontology sets out what would seem to be a category at the very heart of metaphysics in general: a formal category of *objects* (entities, units, individuals or things).

For views of a Newtonian genre, the physical world is first and foremost one of spatio-temporally discrete and separate massive bodies. There are multitudes of planets, countless multitudes, perhaps, of discrete concrete objects altogether generally, having seemingly unlimited ranges of magnitudes—galactic clusters and on up, through grains of dust and boulders, to quarks, neutrinos, and on down. For the Newtonian, any such objects would be in relative or absolute motion, motions themselves following precise and determinate

'mechanical' laws—laws universally applicable to masses altogether generally, simply as containing separate and distinct centres of gravity, regardless of the distinctive features of each one of the multitudes of diverse kinds. Other than 'communication of motion by impulse' as an explanatory principle, the mechanical philosophy itself allows for nothing else. Indeed, Locke, as Newton's disciple, tells us that impulse is "the only way which we can conceive Bodies operate in" [8]. Causal change is reflected in nothing but discrete *events*—the Newtonian impact of one representative billiard ball upon another.

While its influence continues to be massive, its tendencies are to be nominalist, reductionist, empiricist, and atomistic. Furthermore, *qua* Newtonian, its ontology is primarily one of material objects, large and small. During the past 80 years or thereabouts, its leading advocate and 'enforcer' has been Willard Quine. Quine's criterion of ontological commitment, so we are told, "has dominated ontological discussion in analytic philosophy since the middle of the 20th century; it deserves to be called the orthodox view." [9]. This, it seems to me, is right; and the canonical text of this orthodoxy is Quine's book *Word and Object*. But the central lesson of the work, I will urge, is that the pursuit of a deeply misconceived program—even such an extremely influential one as this—can nevertheless lead to positive results. Thanks to the preceding work of logicians (Frege, Russell, Wittgenstein) and linguists (Otto Jespersen), Quine's logico/ontological program points directly to convincing arguments for a logic and ontology that is effectively the complete opposite of Quine's. In this way, the metaphysics of modernity seems to point towards its own destruction.

Now I have spoken of the idea of a scheme of *metaphysical* categories. However, the idea that there might be a further, distinct albeit formally related scheme of categories, within which much of chemical theory is written—a scheme involving an ontologically distinct domain, along with a correspondingly distinct form of causality—is certainly less familiar. And it is just this that I here venture to propose. Chemistry might perhaps be said to be the science of the varieties of matter, their interactions and transformations, in general and as such (although not, indeed, the science of *stuff* in general and as such), but it is far from obvious that the abstract categories effectively deployed within chemistry can be subordinated to those of classical mechanics, those of *object* and *event*. On the contrary, my general view is that the kind of scheme commonly invoked in speaking of the category of separate and discrete bodies—the kind of scheme appropriate to early modern science, and centrally, Newtonian mechanics—is well suited neither to grasping the everyday phenomena and concepts of matter, nor, by the very same token, to grasping the general character or nature of the relatively recent science of chemistry, as a system of quasi-laws and precise formulae concerning substances—the roles of sub-atomic, atomic, and molecular theory notwithstanding. At the same time, the human conceptual scheme that is described by this scheme of categories, at least by intent, has, of course, remained almost entirely unchanged.

The concept of a substance, in this everyday sense, not only occupies the central focus of this work, but its place in logic and ontology frames the questions. Furthermore, the notion is itself neither a notion derived from chemistry in particular, nor from science in general. Rather, it is a notion present in everyday thought and talk from time immemorial (again, talk of *water* in particular must be as old as *Sapiens* themselves).

Yet it is only to be expected that the need for a certain expansion (or perhaps mere re-adjustments) of the structure of the scheme would occur from time to time; and such now would seem to be the case concerning the place of matter in the scheme. Arguably, matter itself escaped scrutiny during the first scientific revolution, thanks to the singular focus of attention during this period upon the heavens—the motions of the planets in relation to the sun—and through the parallel work upon the trajectories of military projectiles, also initiated chiefly by the brilliant work of Galileo, and largely terminated by the astonishing achievements of Isaac Newton. Nevertheless, Newton's companion and self-styled 'under-labourer' John Locke pronounced judgment on the putative kinds with which he was familiar—indeed, the point is emphasized by empiricists since at least the time of Locke, who speaks of the names of kinds as expressions of 'abstract general ideas'—suggesting not

so much abstract *objects*, but rather, the merest *abstractions*. At the same time, however, a defender of the idea of natural kinds might argue that from the standpoint of their scientific status, the putative biological kinds (historically indeed viewed as paradigmatic), are no more than a range of very poor examples, and it hardly follows that the entire idea of a natural kind is thereby open to question. I am certainly sympathetic to this view.[11]

Thus, there is here a certain class of natural kinds whose existence as objectively real and robust kinds is far more difficult to question. The range of *substances*—kinds of stuff or matter in much the chemist's relatively narrow sense—are among the basic chemical constituents of reality, the 'underlying stuff' of which everyday material objects are composed; and within philosophy, their metaphysical status is far from being well understood. From the standpoint of identity, the substances of chemistry are special, and it is this special status that I am hoping to better understand.

So far as the modern philosophical consciousness is concerned, awareness of the fact that a distinct and *sui generis* category of material stuff or matter actually *exists* in the rarified domain of metaphysics, with its highly general, abstract categories, has only just begun to dawn, it would seem, over the past hundred years or thereabouts—thanks chiefly to the work of Lavoisier and, following Lavoisier, the work of Mendelayev—whereby the existence of a class of substances has impressed itself upon the human consciousness. The scope of this class has no doubt expanded with the steady growth of knowledge; but the overall scope of the notion is already clear enough to J.S. Mill, when he writes:

> the possibility... of an ideally perfect Nomenclature is probably confined to the one case in which we are happily in possession of something approaching to it–the Nomenclature of Elementary Chemistry. The substances with which chemistry is conversant, are Kinds [10].

Major attention came to be focused upon matter, in the form of the substances of chemistry and their constitution, only as recently as the late 19th century—the beginnings, in effect, of the second phase or stage of revolutionary scientific activity, out of which emerged a firm recognition of the existence of atoms and molecules, and perhaps above all, the robust theory or theories of quantum mechanics and the problems thereby posed for a mechanistic understanding of the cosmos. The irony in these developments has been that, thanks chiefly to the rise of 'atomism', the category of matter and the concept of a substance in this everyday sense have once again escaped direct scrutiny, remaining largely unexplored—the idea perhaps being that all that was known and all there was a *need* to know concerned the constitution or composition of matter, such that the concept itself again escaped significant examination. At around this time, however, there were developments in both logic and linguistics. On the one hand was the emergence in the late 19th century of modern logic, pre-eminently in the form of Gottlob Frege's *Begriffsschrift*—in the hands of Russell and Whitehead, subsequently to become the prevailing form of logic, and canonized by Willard Quine. On the other hand, the early 20th century is marked by the publication by the distinguished Danish linguist Otto Jespersen of his *Philosophy of Grammar* [11].

### 2.1. Issues in Metaphysics

The further and metaphysical dimension of distinguishing words for substances from other 'unbounded' nouns consists in the main of acknowledging two factors—what I will call their empirically robust criteria of *identity*—along with what I will call their '*cosmic* presence', whether actual or potential. We are aware today, of course, that from a cosmic standpoint, certain substances are indeed widespread and contribute in diverse ways, physical as well as chemical, both to the workings, and to our understanding, of the cosmos. Perhaps the most obvious and relevant quasi-metaphysical fact about *water*—to take the most obvious example—is precisely that water is ubiquitous—it is widely 'scattered' or distributed in some good sense—a point indeed remarked by Quine [7].

But now, what exactly *is* such a thing as a chemical substance? To what metaphysical or ontic category does a substance such as water belong? In very general terms, the answer to these questions seems clear enough. Substances cannot but belong to the category of

universals, and that in an entirely traditional sense. In a nutshell, to say this is to say that the very same substance (the very same liquid, the very same metal, compound, etc.) can occur in any number of disparate regions of space and time.[12] It seems to be both meaningful and true to say that one and the same liquid, namely water, occurs in lakes and rivers all around the globe—these things all have water in them—and again that one and the same precious metal, namely gold, is mined in both Russia and South Africa.[13] The very same substance—the very same metal, liquid, organic compound, and so forth—may be said to be present in any number of places at the same time. It appears to be quite literally true to say that two distinct glasses can contain numerically the selfsame (identical) liquid. A liquid itself, *qua* kind of stuff, can be said to be present wherever there is liquid stuff *of* this kind; and its presence, or existence, does not consist in that of individual concrete 'liquids'. Any containers that contain water can be truly and quite literally said to contain the very *same* liquid; and there is no natural-language sense of 'liquid' for which, if there is water in distinct containers, the liquid in one of the containers could also be said to be a *different* liquid from the liquid in another.

Two glasses of water may contain different amounts of the liquid, and the water in those glasses may be more or less pure and may vary in temperature from glass to glass; but these features are extraneous and irrelevant to the identification or indeed the identity of the liquid itself. Its characteristics *qua* water are independent of any such impurities and environmentally conditioned properties. To say that two glasses of water contain the very *same* liquid (or equally, that two chunks of metal are chunks of the *same* metal or metallic isotope), is to say that the very same set of qualities and powers—in effect, the qualities and powers of the substance *as such*, its essential qualities and powers—are manifest or present in the contents of the glasses. It is quite simply a necessary condition of distinct samples counting as samples of the same chemical substance that there is a certain determinate set of properties that all those samples do in fact possess. Wherever the substance occurs, there cannot fail to be a certain qualitative identity of properties.

There is one problem about this, though. Unlike attributes or properties, which are sometimes (on empiricist grounds) thought of as 'abstractions' from the concrete particular, these universals would seem to be *concrete*. In what follows, I propose to urge not only that chemical substances are indeed universals but also that these universals themselves physically exist—that is, *can* exist, do not *necessarily* exist—in space and time. It so happens that gold, the precious metal, does exist—indeed there is some of *that very metal* in my wedding ring—but it might not; the essence or 'nature' of the substance as such is given purely by its formula.[14] Given that water is indeed a particular concrete substance, a particular compound, thing or object of a rather special kind, being one among the many compounds of basic elements—it does not seem completely unreasonable to describe it as a 'single sprawling object'.

### 2.2. Substances, Identity-Criteria and Laws of Nature

Now it is, among other things, just a plain fact that in the composition of matter (as in much else), there is a certain latent, underlying order to the world—dimly perceived and understood by the ancients with their postulated elements and dramatically revealed by Lavoisier, thanks in part to technological developments, as much as 2500 years later. This underlying order is both physical and chemical; and without some such deeper order, there could hardly be an intelligible and predictable, law-abiding world.[15] Today, such substances as carbon, methane, plutonium, and so forth loom large in novel ways; but from an abstract conceptual standpoint, these are only additions to an existing category; they do not change the ancient principle.

As such, this category of substances cannot be *identified* with the class of actual elements and compounds; these are surely not the only possible forms taken by this category. The number of distinct kinds of stuff, along with their division into elements and compounds, would both seem to be contingent matters of fact. Indeed, it is not easy to see why the category could not be compatible with the existence of substances lacking any kind of

particulate micro-structure. The question concerns the formal character of the very general category or concept of a pure substance—a category that I take the actual class of elements and compounds to instantiate. The formal character of the category of substances in this narrow sense is not an issue for the philosophy of empirical chemistry; it is more general, more abstract and an essentially philosophical issue; it might be characterised as a category (and perhaps the central category) of *metachemistry*. What is theoretically important about the elements and compounds, for current purposes, is that along with water—and crucially unlike mixtures—they all have precisely defined *identity-criteria*, and thereby also modes of behaviour that are fundamentally law-like. What is required, in order for something to count as the *very same* stuff or substance in this sense of the term, is that it has the very same set of both physical and chemical properties, here, there, or anywhere, wherever it occurs. Indeed, what distinguishes chemical substances—elements and compounds—from substances or kinds of stuff in any more inclusive or less formal sense is precisely this.[16] Naturally bracketing impurities, these are the forms of matter that have, in the nature of the case, invariant identifying properties.

With mixtures on the other hand, the numbers and proportions of the various ingredients can vary indefinitely, such that there is no unique, precise and determinate set of properties both common to and distinctive of all samples of, for instance, gin and tonic, clay, bronze, concrete, rye whisky and so on. In consequence, only if the notion of a kind is simply stripped of determinate identity-criteria, can mixtures count as discrete *kinds* of matter.[17]

There exists, then, a certain range of determinate kinds of stuff—a few such as water, gold, salt and iron, familiar from ancient times, many others more recently identified—whereby each of these particular kinds has a specific qualitative identity, given by a unique, precise and determinate set of attributes or properties (including powers)—properties that are, in their very nature, the properties of substances. The physical properties of substances, including such properties as solubility, melting point, boiling point, and density, are essentially *bulk* properties that imply or presuppose concrete three-dimensionality, the filling or occupancy of space. Their chemical properties, on the other hand, seem chiefly relational—powers of lawlike *interaction* with other substances, powers or capacities to undergo time-consuming transformations over intervals of time. Furthermore, such properties are typically independent of the amounts (though not of the proportions) of the substances in question.[18]

But just how could this apparent concreteness of the universal possibly *be?* The answer seems to be this: The concrete reality of the substance, one and the same metal, acid, compound, liquid, etc., seems to be one side of a coin, the other side of which is precisely the absence of any individual 'instances' or natural 'units'. It seems to be precisely *because* a word like 'water' does not *individuate* what it applies to, or divide its reference into *this*, *that*, and *the other*, that it cannot be counted as an abstraction. There is no basis on which to perform the abstraction. In effect, it is implicit in the very idea of a uniform homogeneity that any such material is thereby qualitatively homogeneous; effectively, the same principle applies (as Quine notes) to uniform colours; every part of a uniformly coloured surface will also be coloured in the same mode.

### 2.3. Words for Substances and Mass-Words

Now any analytic investigation of the ontology of substances must proceed chiefly through close investigation of the semantics of the corresponding nouns; and I will be urging in due course that the concept of a *process* is itself important to understanding their semantics. But it is clear that words for substances belong, in a quite distinctive way, to the overall semantic category of mass-words. More precisely, words for substances (that is, words used as the *names* of substances) constitute the generic occurrences of that subset of mass nouns that may be called 'substance-words'. As Quine among others, has noted, the same terms also occur as common nouns or predicates; here I call the former *generic* substance-nouns, and their predicative counterparts *predicative* substance-nouns.

Mass-words are evidently a far more extensive class than the class of substance-nouns, They are also an extremely complex class of nouns, and my initial focus concerns only one specific sub-set of this class—those mass-words that are used, among other things, as the names of *substances*, in the everyday sense.

Plainly, any attempt to explicate the category of substances cannot but involve consideration of the more inclusive group of nouns, and there are general lessons concerning the semantics of mass-words, that are directly applicable to substance nouns as such. Now the semantic category of mass-words, Quine tells us, is 'ill-fitting the dichotomy into singular and general' [4]. The claim, if true, is of the utmost importance. As philosophical dichotomies go, the dichotomy into singular and general might seem about as basic as it gets. That Quine should be concerned about 'the problem of mass nouns', as Davidson once called it, is then by no means surprising. To a first approximation, the dichotomy in question is at one and the same time the dichotomy into universal and particular, abstract and concrete. Little wonder that Quine returns to the issue time and again—and yet proposes only, by his own admission, an 'artificial' resolution. My own view is that Quine is right to be concerned: mass-words are indeed ill-fitting this dichotomy, and any satisfactory resolution of the issue can hardly fail to be disruptive of the entrenched and widely respected corresponding doctrines.

At the same time, Quine's own response to this conundrum is complex and, I find, obscure. He suggests, although does not argue, that mass-words belong, in effect, to what is for us in our 'cognitive maturity' an ultimately unintelligible alien language—a relatively primitive stage in cognitive development that precedes the mastery of individuation, elements of which can nevertheless be captured by holding firm to the dichotomy into singular and general, and adopting an admittedly artificial reductive strategy [4]. In what follows, I take up what seem to me to be Quine's positive insights into the logico-semantical and metaphysical aspects of mass-words, while rejecting Quine's pessimism about the ultimate intelligibility of the phenomena at hand.

The mere formal recognition of this general semantic category appears to be remarkably recent.

### 3. Concepts and Semantics

*The chief semantic contrast*. This work contrasts two semantic categories—those of concrete, so-called (by Jespersen) 'mass-words' and 'thing-words'—with the pair of ontic categories, or highly general concepts, which these two groups of nouns, suitably restricted, express. These underlying concepts I am calling the *object-concept* and the *matter* (or *substance*)-*concept*. Typically, the problems in this area are seen as chiefly problems about mass-words, and perhaps the matter-concept; the object-concept, in contrast, is regarded as well understood, and substantially encoded in our standard formal logic. But this conception of the situation, I would suggest, is itself mistaken: the categories of mass- and thing-words, although mutually exclusive, are *internally related*, and the difficulties over mass-words stem in the main from difficulties over thing-words—and especially, from problems with the underlying ontic concept—that of objects, units, individuals or things, which this range of nouns express. The central problem is not one of mass-words, and Frege's question in particular—the question of 'what we are here calling an object'—is by no means, I think, well put.

Concepts are distinct from their semantical embodiments, and insufficient scrutiny is paid to the diversity of forms in which the object-concept is linguistically embodied or expressed. In fact, the range of sentence-types that standard logic recognises is limited in ways that effectively block access to both the matter-concept and the object-concept equally. The fact that singularity is the sole semantic category of the standard predicate calculus imposes a constraint that is simply incompatible with formalising sentences involving non-singular reference or predication. And unlike the object-concept, the matter-concept represents a category that thus completely falls outside the scope of logic, in the form in which the latter is at present understood.

One central feature of the modern 'concept-script' consists in its theoretical rejection of the Aristotelian 'subject-predicate' sentential form. On the replacement model of language that comes to us from Frege, sentences are instead divided into two broad categories. On the one hand, are fully-fledged, directly referential sentences that exemplify a very different version of the subject/predicate model—in semantic terms, the object/concept model—and on the other hand, are quantified sentences that do not but are constructed on the basis of the former group. This second group is indirectly referential; bound variables taking directly referential expressions as their substituends.

Now I think it goes without saying that general concepts as such and in themselves are never referential; reference enters into their use or application in constructing and perhaps uttering tokens of sentence-types. And an understanding of the concepts here at issue calls for recognition of a further, very different sentence-group. This group, involving bare occurrences of mass-words and thing-words equally, consists of existentially committed sentences that are bare and so unquantized[19]—neither indirectly nor directly referential. Sentences of this non-referential, 'feature-placing' type are, I argue, the most basic form in which the matter-concept and the object-concept are linguistically expressed. For the object concept, this form is ontically, if not semantically, supervenient on the possibility of references to things. But for the central classes of the matter-concept, the possibility of underlying references to stuff or matter seems deeply problematic. An understanding of these sentences calls, among other things, for understanding predications or verb-phrases which are both essentially non-singular and involve the lexically progressive aspect.

An adequate grasp of the object-concept must then include, *inter alia*, a grasp of its potential for linguistic embodiment, or what comes to the same thing, an account of the nature of its relationships to its semantically diverse *embodied forms*. Very often, the concept is employed in identifying or making reference to some one particular individual; but the concept is also employed in speaking in general of objects or one or another kind—'Pigs are sensitive creatures', for example; and again, in speaking of *some* or *all* things of some particular kind; and so on.

Now it is inevitable that mass-words display less diversity in their occurrences than do thing-words. As general terms, they display no indication of semantical numericality; there appear to be no occurrences that differ in the manner of 'woman' and 'women'. At the same time, like thing-words, they appear in contexts in which there is demonstrative reference: One might speak in a quasi-singular manner of *this* or *that* water, rather as one might speak of *this* or that dog; but again, there is no obviously plural contrast akin to that of *these* or those dogs. At the same time, the fact that there is a 'quasi-plural' parallel between speaking of this water as *some* water, and speaking of these dogs as *some* dogs, seems obvious enough. Furthermore, much as there may be determinerless or 'bare' talk of *dogs* roaming the streets, so too there may be bare talk of *water* pouring into our basements. However, the descriptions of these behaviours as 'quasi-singular' and 'quasi-plural' are somewhat obscure—just what would such terminology amount to?

### 3.1. Logical Stratification of Embodiments of Concepts: Pure Embodiments, Determinacy and the Indeterminate

The object-concept is numerically neutral or indeterminate, but referential phrases of the form 'this object' and 'those objects' are plainly not. Just so, the matter-concept is quantitatively neutral or indeterminate, but reference of the form 'this matter' or 'that stuff' is not. 'These objects' and 'some objects' introduce the idea of some determinate but unspecified *number* of objects. An NP (noun phrase) of this referential form has the potential to denote any particular, determinate number of things of a specified kind—a number that is plainly adventitious, from the standpoint of the concept or the kind of thing itself. Likewise, 'this matter' and 'some matter' introduce the idea of some determinate but unspecified amount or quantity of matter. NPs having these referential forms involve a fusion or combination of ontic and matching quantitative concepts. The idea of a number of objects—the correlate, so to say, of the category of plural or collective reference, definite

or not—fuses the ontic category of objects with the quantitative category of number. Just so, the idea of an amount or quantity of matter fuses the ontic category of matter with the quantitative category of non-count reference. Hence the question of just what categories the phrases 'a number of objects' and 'an amount of matter' themselves express or represent must receive the answer that they represent not special kinds of objects (such as 'sets' or 'sums' and 'quantities') but complex hybrid categories, categories expressing fusions of ontic categories with the twin semantic categories of non-singular reference. And in thus expressing hybrid, ontico-semantic categories, the overall ideas lack metaphysical significance or ontic standing.

Basic to the argument is the principle that the semantics of bare nouns do not (or do not essentially) involve the concept of numerical identity. Otherwise put, the use of a bare noun need not involve determinacy; in a sense requiring explication, it need not be referential. To say this is not to say that mass nouns in particular do not accept a concept of identity, for it seems quite clear that they do. However, the introduction of the concept of identity, which is inevitably associated with the phenomenon of reference, introduces something in addition, over and above the involvement of a mass noun, or the matter-concept, in assertions of existence. We need to be able to understand such commonplace forms of sentences as:

There is blood all over the floor,

Water completely surrounds the city centre,

Oil is pouring over the barriers,

And

Smoke is billowing from the chimneys.

And these are precisely sentences that do not essentially involve a concept of numerical identity. However, to clear the ground for understanding them, we do our best to first look at the corresponding and less unfamiliar class of sentences involving count nouns. To this end, I take a simple distributive sentence, 'Dogs are barking', as the reference point; and a sentence such as this is best approached, in turn, through its relationship to cognate non-bare sentences. Consider then the three NL (natural language) sentences:

(a) A dog is barking

(b) Some dogs are barking

(c) Dogs are barking.

(a) and (b), I suggest, are both numerically determinate and indefinitely referential, whereas (c) is not. The identity-related contrast of the bare and non-bare sentences is strikingly evident in discourse-anaphoric contexts. Contrast first the implications, if (a) is coupled with or followed by:

(d) It has been barking all night,

And if (b) and (c) are likewise coupled with

(e) They have been barking all night.[20]

In the case of (a) and (b), the implication in both cases is that the *very same dogs* have been barking all night. But in the case of (c), which is also open to a discourse-anaphoric link, the primary and most obvious reading is one that carries no such implication. All that is implied is that *dogs* have been barking all night; and the truth-conditions of this require only that during any suitably brief interval of the night, at least one dog was barking. In other words, the anaphor in the cases of (a) and (b) is identity-involving, and it is precisely this that is implied in the claim that in their most natural readings, these sentences, unlike (c), are indefinitely referential.[21] For just this reason, (a) in particular is semantically distinct from its truth-conditional equivalent in the predicate calculus. The truth-conditions of (a) are indeed given by its quantified equivalent, the indirectly referential:

(a′) At least one dog is barking

Or

$(\exists x)(Dx \,\&\, Bx)$.

But, I am suggesting, the semantics of the singular NL indefinite article 'a' or 'an' involve something more and carry an identity-involving, numerically determinate or

referential *force*, a force which their quantified equivalent does not, the latter being instead numerically neutral, and supplying no information whatsoever regarding numbers.[22] If the same discourse-anaphoric procedure is followed for the case of the quantified equivalent of (a), that is, for:

(a′) At least one dog is barking,

then a similar discourse-anaphoric identity-involving link is evidently ruled out, and not only on semantic but also on purely syntactic grounds.

In contrast, and in terms of its semantic content, (c) is exactly on a par with (a′), so that truth-conditionally, (a) and (c) are equivalent. The difference is that (c) is, and (a) is not, a purely existential sentence, completely lacking in numerical determinacy, referential force or implications of identity. From the standpoint of truth-conditions alone, (a), (c), and (a′) in particular are all equivalent. None is quantified in the sense of delimiting or specifying quantity or number. If, as with (c), there are said to be barking dogs, then there must of course be at least one (or, in other words, one or more) for there to be any at all. But like (a′), the statement itself supplies no information whatsoever regarding numbers; like the object-concept itself, it is numerically neutral.

Notice here that (b) is distinct from (c) in both of these respects. That is, to say that *some* objects are thus-and-so is to say that a certain determinate *number* of objects are thus-and-so; to say that *objects* are thus-and-so is not—it is, again, numerically neutral. The truth-conditions of (b) are given by:

(b′) At least two dogs are barking.

In other words, (b) is equivalent to 'A number of dogs were barking all night'; and the idea of a number of objects is essentially the idea of determinate multiplicity; one thing alone is not enough to count.[23] Contrary to this view, it has been claimed that the truth-conditions of sentences such as (c)—'Dogs are barking'—are not distinct from those of (b). Eytan Zweig, for example, opines that, like *non*-bare plural sentences, bare plural sentences require *at least* two objects for their truth, even though—as he somewhat obscurely remarks—'the plural noun itself does *not* assert more than one...' [17]. Much of course depends upon the nature of the predicate; but the plural form of (c) notwithstanding, it is *not* rendered false if there is just one object satisfying the description. The statement 'There are snipers in this area', for instance, would not be judged false, if there were a lone sniper in the area. It is possible that one who utters the sentence believes that there are or might be a number of snipers in the area; but this has no impact on the truth of what they (actually) say. The sense that bare plural sentences in general require at least two objects for their truth is not simply a consequence of their plurality but seems largely the result of pragmatic rather than semantic factors. The fact is that (c) counts as false, just in case no dogs are barking; and:

There are dogs in the garden

counts as false, just in case there are no dogs in the garden; otherwise, it is true.

Compare this situation now with a parallel non-count case. Lacking singularity, there are just two forms of non-count sentences that are not definitely referential. As examples, I propose the following. On the one hand,

(f) Some water is on sale at the well

is a derivative or secondary sentence, while the fundamental or primary sentence is

(g) Water is on sale at the well.

In relation to anaphora and identity-potential, these match the corresponding thing-word sentences: (f) accepts *bona fide* anaphora, whereas (g) does not. With (f) there is, for instance,

(f′) It's been on sale all week; no-one wants it.

And for (g) there is

(g′) It's been on sale all week; everyone is buying it.

The relationships of (f) and (g) are parallel to those of (b) and (c): (f) presupposes (g), but not vice versa. Like (b), (f) implies determinate (or limited) but unspecified quantity—there may be just a litre left—whereas (g) is open-ended and wholly neutral, indeterminate,

lacking any built-in limits, and in just this sense, is limitless in quantity.[24] And like (g), a Strawsonian feature-placing statement such as

(h) There is water for sale at the well

is both existential and unlimited, i.e., unquantified. Here the category of stuff itself appears directly, unmarked and unmodified by the presupposition of quantitative elements, identity or potential reference; and seen in this light, Strawson's claims about his feature-placing sentences do not appear outrageous. In contrast, the d-type sentence has quantity and referential-use potential added on. Both (b) and (f) involve a general concept along with quantitative limitation. 'Some' marks an element of empirical information, adventitious from the standpoint of the ontic categories themselves. These may be called 'impure' or hybrid ontico/semantic forms, forms that lay the semantic basis for non-singular identity-statements. In contrast, the form of (g) represents the pure or subjectless embodiment of the matter-concept.

We may now sum up these observations. In the first place, unlike its non-bare counterparts, the bare sentence

(e') Dogs have been barking all night

is numerically indeterminate and is not indefinitely referential. There is a straightforward sense in which, as the sentence stands, it has a temporal, aspect-involving predicate which need be true of *nothing*: There need be no dog which has been barking all night. And in this sense, although it has—or better, perhaps, can be reduced to—an indirectly referential truth-conditionally equivalent sentence, the sentence has a grammatical subject that corresponds to no logical subject. As it stands, *qua* plural, it lacks a directly or indirectly referential subject; it has no non-grammatical subject/s. It is only by changing the structure of the grammatical predicate that it is reducible to an indirectly referential or quantified singular sentence, involving quantification over dogs and times, and this is possible because the predicate is distributive. The fact remains then that the semantically plural structure is no idle feature of the bare sentence form—were 'barking' replaced by 'gathering', or 'milling about', there could be no such reduction. And secondly, sentences with the form of (c), (e) and (e') are identity-compatible: While they are not identity-involving, it is evident that their truth does not require a non-identity through time. Syntax notwithstanding, (c) is numerically neutral and also non-referential; here, there is 'undifferentiated' or numerically indeterminate talk of *objects*. On the other hand, in (a), there is determinate talk of an *individual* object, and in (b), there is determinate talk of a *number* of objects. Neither mode of embodiment (*mem*) is numerically neutral; both are identity-involving.

(c) is a semantically distinctive *mem* which provides an existentially committed neutral way of speaking of *objects of a kind*. In contrast, in indicating the involvement of a number of individuals and laying a semantic basis for plural identity-statements, the presence of the indefinite non-singular determiner 'some' constitutes the introduction of an element that is adventitious from the standpoint of the kind or ontic category itself. The information that there are a certain number of things of a certain kind that are thus-and-so is taxonomically, hence ontically irrelevant. *Pace* Quine, the sole categorially or ontically salient fact, consists simply in the information that there are *things of the kind* that are thus-and-so, in a given context—and there is, of course, a parallel contrast for the case of mass-words. The amount of stuff that there may be, of whatever kind or kinds, is no part of the ontology; what matters categorially or ontically is just that there is *stuff of one or another kind*. This is a more basic form of thought than those involving 'some', identity, or reference, and which introduce amounts; and sentences having this form may be appropriately described as pure embodiments of the corresponding concepts (See Figure 1 below).

| OCCURRENCE TYPES: | GENERIC | existentially significant = | | 'instantiation' = | embodiment of concepts |
|---|---|---|---|---|---|
| | *Non-existential Category / Concept / Kind* | *Existential* | | | *Referential* *definite* |
| | | *ontic / neutral unquantified / indeterminate* | *non-neutral / quantified / indefinite* | | |
| bounded thing-words | 'Objects' | 'objects' | 'some objects' 'an object' | | 'these objects' 'this object' |
| unbounded mass-words | 'Stuff' | 'stuff' | 'some stuff' | | 'this stuff' |

**Figure 1. Overall Taxonomy**.

*3.2. Two Questions*

There is the question of exactly what this lack of identity-involvement has to do with the nature of the criterion of ontic commitment for matter; and there is the question of what its actual physical significance is—what implications for our thought of matter does this lack of built-in identity have?

We have so far remarked that the very same concepts—including the object-concept itself, in general and as such—can be embodied in a semantical diversity of ways, including, among other things, four distinct varieties of reference—both definitely and indefinitely referential singular and plural terms. All these modes of embodiment or *mems* represent forms of embodiment that are in one way or another *determinate*—referential or identity-involving. There is, however, a further and, I think, more fundamental mode in which the object concept is embodied—a *mem* that wholly lacks this feature of determinacy, and thereby sets no boundaries on the application of the object-concept. Unlike (a) and (b), it seems clear that not all talk of objects *is* determinate or referential in the first place, even in the minimal *indirect* sense of involving the use of bound variables. Thing-words can occur in sentences, true or false, without determiners, boundless and 'bare'; and here, the corresponding concepts can have application, even in the absence of arguments. Corresponding to this, there is a somewhat neglected class of concrete, non-generic sentences, over and above the 'standard' group of indirectly and directly referential sentences. This class is the class of general, existentially committed, non-singular sentences, the distinctive feature of which is that they incorporate *bare nouns*. Among linguists in particular, bare nouns are sometimes characterised as *unbounded*, and what this is chiefly (and, I think, correctly) understood to mean is that such sentences are numerically non-determinate, or non-specific as to quantity. Unsurprisingly, the feature has implications for the concept of identity—but also, and especially, for the matter of ontological commitment. Some such sentences are syntactically of traditional grammatically subject/predicate, NP/VP form, as with 'Bodies covered the ground' and 'Ants are pouring from the cracks'. Others are explicitly existential, thus 'There were bodies all over the floor' and 'There are ants here'. These bare assertions of existence are not *indefinitely* referential; as we will see, it seems clear

that they are not identity-involving, but further, I will suggest, they need not be *indirectly* referential either. Compare now the bare sentence:

    (c) Dogs are barking,

as a distinctive *mem* of the object-concept, with the indefinitely referential sentences (a) and (b). And consider in particular their differential relationships to the object-concept itself. In terms of truth-conditional content alone, (c) is exactly on a par with (a)—and also, of course, with the natural-language counterpart of the formal existentially quantified sentence (4)—that is, with (3). The key difference is that unlike (a), (c) is what I will call a *purely* existential sentence—entirely lacking in numerical determinacy, referential force or potential for the involvement of identity. Truth-conditions notwithstanding, both sentences (a) and (b) are indefinitely referential, identity-involving or numerically determinate, in a sense in which (c) is clearly not. Unlike (a) and (b), (c) has no potential for *bona fide* cross-reference or anaphora; it lacks identity-involving force. If (c) is followed by or conjoined with (e), the coupling carries no implication that the *same* dogs have been barking all night; it is here sufficient that at any time during the night, at least one dog was barking. Manifestly, there need be *nothing*, no subject, of which the predicate of the unreconstructed sentence:

    (e) Dogs have been barking all night

is true; there need be no dogs that have been barking all night (although, of course, the statement is perfectly consistent with that possibility).

While (c) has existential import, then, it is not quantified *in the sense of* delimiting or specifying the number of dogs in any way; its truth is compatible with the existence of any number of barking dogs whatsoever. If dogs are said to be barking, then there must, of course, be at least one—in other words, one or more—for there to be any at all. But the statement supplies no numerical information whatsoever. Unlike (a) and (b), yet like the object-concept itself, (c) is numerically neutral or indeterminate: It presupposes no conception of a *limitation* on the range of application of the relevant concept. And because it is a purely existential sentence—neutral, and, in this sense, unquantified—it may be said to *directly* embody the concept, in its pure and pristine form. (c), therefore, of the three, is the more fundamental form of sentence. It is truth-conditionally *presupposed* by (a) and (b). Indeed, I will suggest, it can in principle be used prior to the ability to count, or even to re-identify individual objects—although to fully grasp the object-concept is to grasp its capacity to figure in all these different forms of sentences.

Only sometimes is the object-concept embodied in determinate subject-introducing, referential contexts: The property of referentiality introduces a complexity above and beyond the pure application of the concept itself. In one form or another, a mechanism of plural *reference* enters the picture, along with the concept of plural identity, precisely when the concept of determinate number, whether specified or not, enters the picture, as in (b). The upshot of this, then, is that among the diverse roles the object-concept plays in talk and thought, there is one particular role that is profoundly *pre-individuative*—independent of the notion of identity. In other words, there is a notion we might dub that of 'indeterminate multiplicity', as against, perhaps, 'determinate plurality', whereby there may be thought or talk of the presence of objects—say, of fish or sheep—with no implication of the *sameness* of a number of fish or sheep, nor indeed of any individual fish or sheep. (And in some contexts, I will suggest, this gives rise to forms of thought or talk that are positively and irreducibly hostile to the concept of identity). Strangely enough, as it may seem, this implies that it is possible to talk and think of sheep and fish, and of their multiplicity, without the actual ability to *identify* (and so to distinguish, and track, or re-identify) individual sheep, whether individually or collectively. *A fortiori* implies that it is possible to talk and think of sheep and fish without the ability to *count*. In what I am supposing is the fully-fledged sense, it involves the use of symbols such as numerals and is deployed in such things as conducting a census, on however small a scale. And although my claims concern the nature of the object-concept in itself, there is nevertheless *evidence* for these differences relating to the concept of a cognitive-psychological and anthropological character.

### 3.3. The Contribution of Otto Jespersen

Around one hundred years ago, Jespersen drew the world's attention to a basic linguistic fact, and one with some potential bearing on major issues in logic and metaphysics. Concrete nouns divide quite naturally, and almost exhaustively, into just two semantic groups, reflecting the two kinds of concepts (and perhaps also the realities) to which these expressions would seem to correspond. They divide, roughly and informally, into terms for concrete objects, individuals or things, and terms for material stuff or matter. In a brief, dense, and unusually penetrating discussion—taking place, ironically, not so long after the appearance of Frege's foundational *Begriffsschrift*—Jespersen observes that there are:

> a great many words which do not call up the idea of some definite thing with a certain shape or precise limits. I call these 'mass-words'; they may be either material, in which case they denote some substance in itself independent of form, such as silver, quicksilver, water, butter, gas, air, etc., or else immaterial, such as. . . satisfaction, admiration, refinement, from verbs, or. . . restlessness, justice, safety, constancy, from adjectives.[25]

No one, in my view, better represents the issues that here confront us than does Jespersen. The two concepts here at issue are expressed by what Jespersen called 'thing-words' and 'mass-words'; I am calling the concepts in question the *object-concept* and the *stuff-concept*.[26] Consequently, this work as a whole has been divided into three main parts. The first and near-introductory part is devoted to the conceptual contrast underlying this linguistic dichotomy. In principle, this itself calls for an account of every distinct type of linguistic embodiment of each of the two concepts associated with these nouns, and in that sense, it calls for an account of every relevant logical form of assertion. Jespersen's comments on these matters seem to me foundational.

It is, I think, by no means accidental that Jespersen is not a philosopher but a linguist. Here, however, I approach Jespersen chiefly from a logico-semantic and conceptual standpoint—in effect, the standpoint of philosophical or conceptual analysis. Jespersen's discussion goes well beyond the scope of linguistics proper.[27] Dramatically alluding to two metaphysically distinct 'worlds' or realms, Jespersen tells us that in an ideal language constructed on purely logical principles, a form that implied neither singular nor plural would. . .be called for when we left "the world of countables (such as houses, horses, days, miles, sounds, words, crimes, plans, mistakes, etc.) and got to the world of uncountables".[28] Jespersen goes on to speak of a 'fundamental logical contrast' between mass-words and thing-words: "Mass-words", he forcefully asserts, "are totally different, logically they are neither singular nor plural, because what they stand for is not countable". In any event, Jespersen clearly suggests, but the trained linguist (and not ontologist) that he is, is hardly in a position to develop a logic and ontology of words for his 'substances' that is utterly distinct from mainstream, object-centred logic and ontology. Jespersen gestures towards the idea of a category that is *not* that of 'definite things with a certain shape or precise limits,' although he does not mention any positive attribute of what, if anything, he supposes it to actually *be.*

However, Jespersen's suggestive, quasi-metaphysical contrast between the idea of 'some definite thing with a certain shape or precise limits', and that of a 'substance independent of form'—in effect, his contrast between the ideas of having built-in boundaries or shape and of *not* having such boundaries or shape—is reflected and presumably grounded in two different sets of subordinate properties—properties that are arguably also basic primary qualities in the familiar sense—just those qualities possessed by 'definite things of a certain shape', on the one hand, and those possessed by kinds of stuff or substances that are 'independent of form', on the other.

Jespersen's quasi-metaphysical contrast between the idea of 'some definite thing with a certain shape or precise limits', and that of a 'substance independent of form'—in effect, his contrast between the ideas of having built-in boundaries or shape and of *not* having such built-in boundaries or shape (formless or unbounded contrast) is arguably reflected

and perhaps grounded in two metaphysically distinct sets of subordinate properties—properties which are also basic primary qualities in the familiar sense—just those generally recognised qualities possessed by 'definite things of a certain shape', on the one hand, and those less widely recognised qualities possessed by kinds of stuff or substances that are 'independent of form', on the other.[29] (Locke has 'figure' (or shape) and 'number' among the qualities of objects; for materials or stuff, we might bring in density, viscosity, boiling and melting points, etc.). Perhaps no less importantly, many 'secondary qualities' parallel the shape/formless contrast. Colours, for example, much like density, specific gravity, etc., are primarily qualities of stuff, not things—the colours, for example, of metals (sharing the names of the substances themselves—'gold', 'silver' and 'copper'), qualities of pigments, dyes, and acrylic, oil, or water colour paints.

Around a mere one century on, Jespersen's farsighted thoughts and proposals on these issues remain lacking any major influence or uptake; indeed, they seem largely unrecognized or unappreciated. Yet, at the end of the day, as it seems to me, Jespersen has opened the door—ironically, perhaps, by way of Quine—to a more inclusive perspective for the logic and ontology of mass-words, and a whole new range of questions for logic and ontology. It is in light of such observations that we are obliged, I think, to address a range of problematic facts about natural languages, in relation to these logical ideas. These are largely verifiable facts resulting from the recognition, early in the 20th century, of the distinctive behaviour of what the Jespersen originally dubbed as 'mass-words'.

*3.4. Jespersen's Major Mistake: Tableau and Comments*

The upshot of these observations is not that mass-words have nothing in common with thing-words—quite the contrary, in fact. For although I am urging that Jespersen's 'neither singular nor plural' judgement is correct, his view of the significance of this fact is surely not. Unfortunately, Jespersen is misled on account of misunderstanding the significance of his own central insight. He plainly intends his 'neither singular nor plural' insight as justifying the further claim that in logical terms, as compared with thing-words, mass-words are (as he puts it) 'totally different'. But this, vague though the expression is, is undoubtedly a major mistake—and one that, post-Jespersen, is often repeated. Indeed, far from justifying Jespersen's sweeping assertion concerning the difference, his two claims seem flatly incompatible, and the second, in particular, is surely false.

It did not seem to occur to Jespersen that in describing certain differences between mass- and thing-words as he rightly did, he was thereby *also* in effect describing certain commonalities—and thereby leading us directly to the logico-semantic form of bulk concepts. Thus, among other things, to be semantically singular is clearly to be *non-plural*, just as to be semantically plural is, among other things, to be *non-singular.* Hence, both types of nouns lend themselves to being related through a single, relatively simple and unitary framework. To say that mass-words are semantically neither singular nor plural is just to say that they are both non-singular and non-plural. But now it is a mere truism that plural occurrences of thing-words are non-singular—and indeed that singular occurrences of thing-words are non-plural. Hence for a mass-word to be neither singular nor plural would not only be for it to differ semantically from, but also for it to have much in common with, occurrences of thing-words—a fact that evidently bears on the understanding of mass-words and thing-words alike. Taking the 'neither singular nor plural' maxim as a serious hypothesis, the linguistic evidence in its favour would seem to be convincing. And on just this basis, it is possible to sketch out an integrated and coherent semantical-cum-conceptual structure, conceived as the semantical embodiments of two highly general categories or concepts, within which the diverse semantical phenomena may be systematically assigned a place.

The basic shape of the relationships between these groups can then, I suggest, be neatly represented in the following tableau (Figure 2). The two sets of nouns, words for things and words for stuff, are *displayed* as not only being differentiated from one another but also being intimately connected, as parts indeed of the very same overall unifying

structure. This structure seems open to both generic singular terms and non-generic general terms; and at the same time, as representing three semantic values, it represents the expression of just two related concepts—that of things or objects, alongside that of stuff. The semantic values attach (of course) to the expressions; whereas *ex hypothesi*, the concepts, not being expressions but rather what is expressed, are concepts *of* putative kinds or categories of *being*—more precisely, concepts that might perhaps, but do not necessarily represent, *bona fide* kinds or categories of being. Again, there is a long-standing question concerning the most appropriate conceptual characterization of the mass-word (in terms, that is, of the concept it embodies, represents or expresses). And here, the brief answer is that these concepts are of course bulk concepts; and furthermore, that such concepts might be said to underlie a peculiar sort of 'fusion' of singular and plural expressions; and the question of how such a feat can be achieved is answered by the fused range of quantifiers and determiners—demonstrating a certain complexity, but crucially because these phenomena to be fused represent different levels of occurrence of the expression. That is, uses themselves must be divided into the more complex, *referential* uses of these expressions, and the simpler, more basic *non-referential* uses of these expressions, a proposal I have developed further.

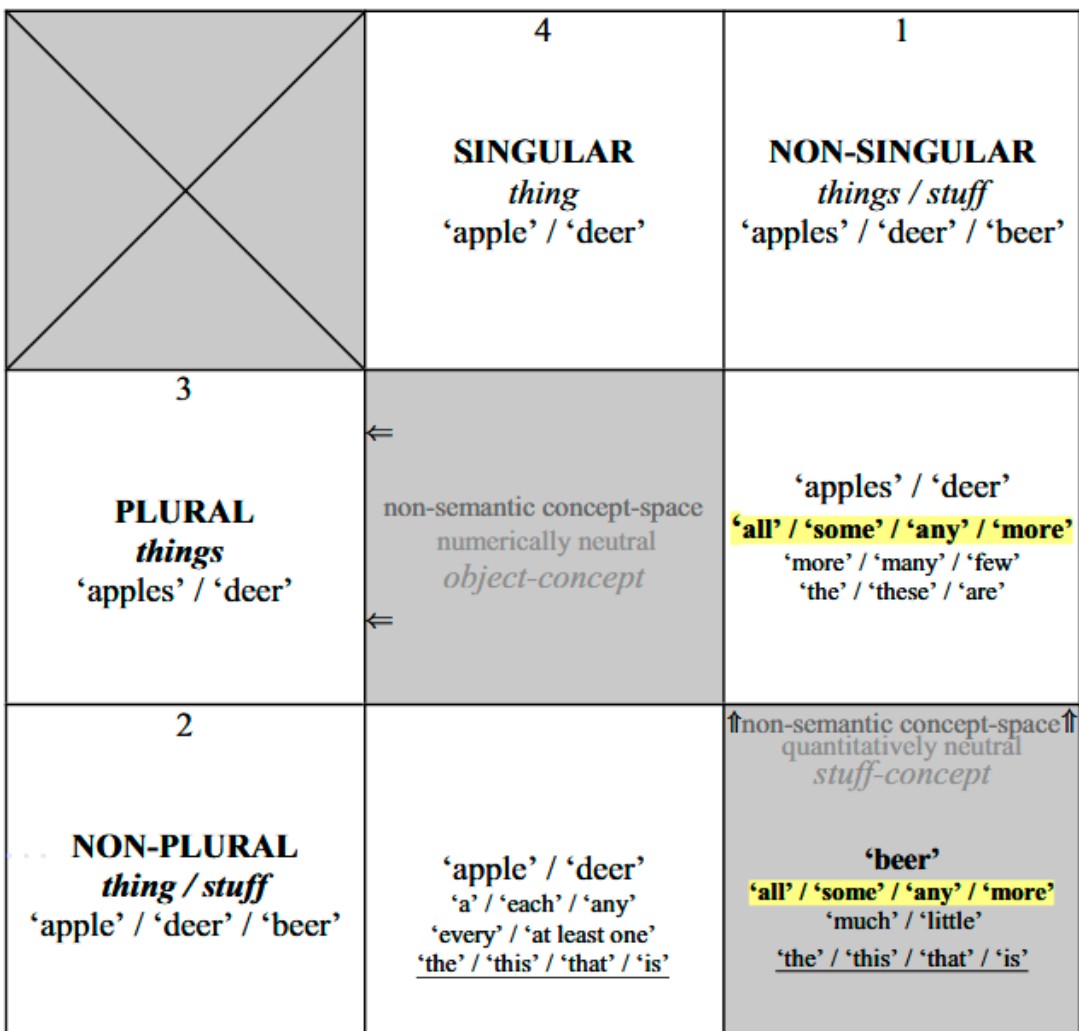

**Figure 2.** Analytical tableau of common nouns: three semantic values, two embodied concepts.

*3.5. Some Comments on the Analytical Tableau*

(i)  A note on the numbering system in the first row and the first column of this tableau, the highest number being in my view the 'source' of lower-number cells and categories, such that at the bottom of the entire heap is, precisely, the singular. There is indeed an argument here, based on matters of simplicity and complexity of the contained concepts, for another time perhaps.

(ii)  There is a bold font emphasis on the shared non-singular quantifiers in cells {1,3} and {1,2}; similarly, the shared non-plural determiners, demonstratives, articles and verbs in cells {4,2} and {1,2} are underlined. Here, the commonality of mass and singular expressions is directly expressed. The tableau both embodies and reinforces Jespersen's 'neither singular nor plural' claim, as well as indicating the depth of the interconnections between these two great semantic categories of nouns.

(iii)  Turning to the non-singular column 1, what I hope to be most clearly displayed here is the *sharing* of what might otherwise be conceived as a range of *plural* quantifiers—but given the fact that they share these quantifiers with (non-singular) *mass*-words, they can only be properly designated as belonging to the more general semantic grouping of *non-singular* quantifiers.

(iv)  While the concepts here are evidently interconnected, they are certainly not symmetrical. There is one conspicuous major asymmetry between the non-singular and non-plural status of mass-words. In sharing with the plural thing-words the capacity to be used in bulk contexts, mass-words are non-singular; but they are non-singular in a different, although related, way.[30] While they share with singular words the aspect of being non-plural, they are also in general *non-atomic*—crucially, there are no semantically determined smallest units.[31]

(v)  The shaded *non-semantic* 'concept-spaces' of cells {4,3} and {1,2} represent the object-concept and the stuff-concept as the underlying 'sources' of the corresponding expressions, the small arrows indicating the relative proximity and importance of the embodied and potentially neutral plural thing-words to the neutral object concept *as well as* to the neutral stuff-concept. The semantic non-singular/non-plural *embodiment* of the stuff-concept is thereby also established—metaphorically 'coinciding' directly with the ideal concept-space of the stuff-concept itself.

(vi)  I have included two examples of thing-words, in one of which number is syntactically marked ('apple'/'apples') and one in which it is not, making this latter kind of expression appear grammatically somewhat closer to the mass-words.

## 4. Unboundedness of Nouns, Determinacy and Indeterminacy of Noun Phrases

*Existence*. Among the most humdrum, and perhaps the very simplest, existentially committed statements involving mass-words are such sentences as

(2a) There is water in the basement,

(2b) Water is leaking from the basement pipes,

(2c) There's mud all over the floor,

(2d) Mud is oozing out of cracks in the wall.

The standard account of existentially committed statements (which focuses, of course, exclusively on statements involving count nouns) is that they are semantically determinate, albeit in the weak sense of being *indirectly* referential, or of being, as they stand, semantically 'incomplete' ('There is something, x, such that it is F', or 'There is at least one x, such that x is F'; and so on). Containing variables, they thereby implicate the possibility either of determinate substituends, making reference to individuals, or simply of determinate values for the variables. In the case of mass-words, the principle would naturally call for the introduction of such assertions as (1a)–(1c), involving the use (in this case) of overt determiners. But considering instead the *prima facie* less complex statements (2a)–(2d), it is far from obvious that these are indirectly referential or semantically 'incomplete', somehow implicating reference to any such determinate 'amounts' or 'quantities'. Far from being incapable of being used independently of quantitative adjuncts, mass-words appear thus

to be perfectly capable of standing on their own; the question is rather a question of how such free-standing assertions might be understood. And here (too), Quine has valuable insights to acknowledge (see Section 3.1). A semantically distinct, non-referential or non-determinate interpretation of these sentences, matching their actual grammatical form, would seem more appropriate, and, at the same time, simpler or more fundamental than (1a) through (1c). For what, if anything, could an identifiable subject for such sentences as (2b) or (2c) possibly be? We shall need to contemplate seriously the possibility that there is indeed *no such thing*; this option is examined and supported in some detail in the sequel.

But there is also a more general point: Naturally enough, general terms display much more flexibility in their behaviours than their cognate singular counterparts. There is a wide variety of contexts in which they can typically appear. Now, the terms corresponding to the underlying object- and stuff-concepts are both non-referential in a weak sense. That is, each is capable of being used as a bare NP, applied independently of determiners and quantitative adjuncts (just so long as the uses are non-singular). However, in a stronger sense, only terms expressing the matter-concept are non-referential, since singular occurrences of count nouns in English standardly require determiners; and whether singular or not, count nouns are in any case directly or indirectly referential or identity-involving. And at one remove, indeterminate plural sentences themselves are associated with identity and determinacy in the singular. With mass-words, however, the phenomenon of reference arises *only* at the level of particular, determinate *uses* or applications of the concept, and fundamentally, in its seeming application to something or other 'in particular'. It is neither a built-in feature of the mass noun or concept in itself nor of all uses of the term, and specifically, not of those uses that are bare. Bare non-singular noun phrases are just those occurrences or uses of mass-words and plural thing-word nouns that involve no use of a determiner (as with 'Water is coming under the door', 'Refugees are flooding across the borders', 'Dogs were barking all night', 'Dust is settling on the furniture', etc.).

And it is hardly insignificant that the post-Fregean notion of a 'predicate', properly so called, does, if only indirectly, involve or implicate the notion of a subject—whereas the more purely linguistic notion of a verb-phrase, I will urge, does not. We have already suggested that Jespersen's talk of not 'having a certain shape' and of not 'having precise limits' can be so construed that the concepts involved are clearly distinct, even though it is also clear that they cannot fail to overlap. Indeed, perhaps the most fundamental weakness of the standard semantic model, SSM, consists in its effective conflation of the two concepts of boundedness and determinacy. This unwitting conflation is the inevitable result of the assumption of semantic singularity, as I consider in more detail in the sequel; but the plain fact is that the uses or occurrences of mass-words—much indeed like those of thing words nouns as well—need not be determinate: This is so, precisely when the uses in question are non-singular, and may for that reason alone be bare—as are both (1) and (2), along with (2a)–(2d), as opposed to uses in determinate sentences, sentences involving indeed determiners, as with (1a)–(1c). This crucial possibility is completely blocked from view, given the assumption that mass-words are semantically singular, since semantic singularity is essentially determinate. On that assumption, the thing words and the mass-words are just two sub-sets within the ontic category of objecthood, rather than being (what I maintain, with Jespersen, they really are) two ontically separate and distinct categories.

Furthermore, ironically, it is just such indeterminate uses of a mass-word that correspond directly to its ontic and metaphysical significance. In other words, it is sentences such as (1), so I will urge that they directly embody or express the ontic content of such a noun. The standard semantic model does not so much fail to recognize the difference between the thing words and the mass-words—no account that acknowledges the semantical distinctness of these groups of nouns can possibly do that—rather, because it fails to grasp the non-singularity of mass-words, it cannot acknowledge the possibility, and *a fortiori* the vital significance, of the essential *indeterminacy* of mass-words themselves.

### 5. Unquantified Assertions of Existence

Quantified assertions of existence would seem, then, to be just one sub-class within the general class of such assertions. For we have seen a further, broader class of concrete, non-generic sentences, over and above the 'standard' pair of indirectly and directly referential sentences. This is the class of general, existentially committed, non-singular sentences, both bounded and unbounded, the distinctive feature of which is that they incorporate bare nouns. Some such sentences are (syntactically, at least) of traditional grammatically subject/predicate, NP/VP form—as with:

Blood covered his face,

Water was dripping from the ceiling

While others are explicitly existential, thus

There was blood all over his face

And

There was water here.

These bare assertions of existence are, of course, not directly referential; but neither are they indirectly referential. Concepts can be applied in a variety of ways, and only sometimes are they used in subject-involving, referential contexts; they can also be semantically embodied in statements that preserve their unquantified, 'pure' or neutral form. I now therefore propose a series of exemplary sentences containing mass-words and process-verb phrases, followed by a parallel series of sentences containing thing words and process-verb phrases. (Evidently, some of the predicates occurring with thing-word nouns are *only* possible for such nouns, in virtue of their boundedness. Hence predicated of a typical mass-word, 'bombarding' is manifestly incoherent). All the following sentences, in virtue of their bare non-singularity, are indeterminate and neutral.

As suggested in the following tableau (Figure 3), it is with the inclusion of such semantically indeterminate sentences, in the class of sentences headed by bare noun phrases, that the ontic role of substance words is most directly expressed. I am assuming here that such generic sentences as 'Water is a compound' are non-ontic, or at best are weakly ontic, since it is by no means clear, from the truth of such a sentence, whether it follows that there *is* any stuff of the relevant kind, or not. The tableau aims first and foremost to capture the use of nouns as general terms and not as singular terms, although in the cases both of unbounded and of bounded nouns in their plural form, these two uses coincide. But there is also a more general point: Naturally enough, general terms display much more flexibility in their behaviours than their cognate singular counterparts. There are a wide variety of contexts in which they can typically appear.

Now, the terms corresponding to the underlying object- and stuff-concepts are both non-referential in a weak sense. That is, each is capable of being used as a bare NP, applied independently of determiners and quantitative adjuncts (just so long as the uses are non-singular). However, in a stronger sense, only terms expressing the matter-concept are non-referential, since singular occurrences of count nouns in English standardly require determiners; and whether singular or not, count nouns are in any case directly or indirectly referential or identity-involving. And at one remove, indeterminate plural sentences themselves are associated with identity and determinacy in the singular. With mass-words, however, the phenomenon of reference arises *only* at the level of particular, determinate *uses* or applications of the concept, and fundamentally, in its seeming application to something or other 'in particular'. It is no built-in feature of the mass noun or concept in itself, nor of all uses of the term, and specifically, not of those uses that are bare. Bare non-singular noun phrases are just those occurrences or uses of mass-words and plural thing-word nouns that involve no use of a determiner (as with 'Water is coming under the door', 'Refugees are flooding across the borders', 'Dogs were barking all night', 'Dust is settling on the furniture', etc.).

I have suggested that the unity or integrity of the object-concept consists precisely in its numerical neutrality. Nevertheless, there is, I think, a strong and perhaps even natural tendency to suppose that the concept is *not* numerically neutral, that it, is on the contrary,

the concept of *an* object (or, in other words, the concept of a *single* individual or unit)—and is, on this account, intrinsically singular, and so in one form or another (if only indirectly) referential, and truth-conditionally represented by the quantifier 'at least one'.[32] This is, I believe, the deepest error in the traditional approach. (The idea that singular sentences constitute the simplest and most fundamental form of assertion, and the belief that speaking of *an object* is, among other things, less complex than speaking of *objects*, come to much the same thing).

| | |
|---|---|
| (a) **Blood** is oozing from his wounds | (a) Minutes after the earthquake **people** were flooding the streets |
| (b) **Blood** was trickling from the corner of her mouth | (b) **Resistance fighters** are besieging the base |
| (c) **Traffic** is flowing smoothly along the 401 | (c) In Bangladesh, the country's top scientists warn of the specter of millions of **climate refugees** streaming into neighbouring India |
| (d) **Oil** is pouring over the barriers | (d) **Foxes** were milling around the hen-house |
| (e) Cheap imported **furniture** flooded into the domestic market | (e) **People** were pouring into the streets, many with weapons |
| (f) **Smoke** is billowing from the chimneys | (f) **Vehicles** are flowing smoothly along Highway 401 |
| (g) **Water** was pouring out of cracks in the dam | (g) **Emails** begging for cash are bombarding inboxes with ever increasing frequency |
| (h) **Mud** began oozing through cracks in the wall | |

**Figure 3.** Bare plural and unbounded Nouns.

With the stuff-concept, on the other hand, the neutrality in question is quantitative. Along with singular thing-words, occurrences of mass-words are semantically non-plural (hence '<u>this</u> water'), and along with plural thing-words, non-singular (hence '<u>some</u> water'), but the concept neutrally expressed by 'water' has no such quantitative value. As such, the stuff-concept is quantitatively neutral; and both stuff- and object-concepts can be embodied in less complex, non-referential forms of existentially committed sentences which are unquantified—semantically indeterminate sentences involving the quantitatively neutral use of bare mass and thing-words. And it is just such sentences that constitute expressions of existence in the philosophically interesting, ontological sense. The answer to the question, 'In what does the existence of *water* (the compound—the familiar *kind* of stuff) actually consist?', is, I suggest, extremely simple—even, superficially perhaps, disappointingly so. The answer is not that it consists in such 'things' as this or that water—supposedly discrete instances, quantities, masses, parcels or amounts of water, 'mereological wholes' or 'parts' of the stuff. The answer, rather, calls for the use of the bare, unquantified noun; its existence consists simply in that of stuff *of* the kind. There is, quite simply, water here, water there—water all over the place. The simplicity of this answer is nonetheless deceptive:

A satisfactory answer must also involve an account of what it is that the question is an answer to—that is, the question of the existence of the *substance*, water. That issue can only be addressed elsewhere. There is, however, one final observation deriving from the comparisons in Figure 3. Plainly, not only stuff, but also things, can *pour*. The difference shows up with such process-expressions as 'to drip' or 'dripping'. Solid objects, *things*, cannot drip; only liquid stuff can drip. The idea that fluid stuff represents a simpler and more basic ontic category is reflected in such facts as these.

**Funding:** This research received no external funding.

**Institutional Review Board Statement:** Not applicable.

**Informed Consent Statement:** Not applicable.

**Data Availability Statement:** No new data were created or analyzed in this study. Data sharing is not applicable to this article.

**Conflicts of Interest:** The author declares no conflict of interest.

## Notes

1　There are of course those who find themselves obliged to question Aristotle's schemes, and in particular the relationship between his *Categories* and his *Metaphysics*. See in particular *Aristotle's two Systems* by Graham [1] in a related way, I hope to show that Quine questions the relationship between the kind of *Categories*-scheme as reflected in the 'canon' and this category of 'stuff' or 'substance' as identified by Jepsersen, in Quine's observation that the semantic category of mass nouns is, as he tells us, 'ill-fitting the dichotomy into singular and general'.

2　We have already remarked a philosophically familiar sense of the expression 'substance', in which such discrete individuals or things as pebbles, planets, dogs and cats count as distinct substances—'substances' so-called now, chiefly in the Aristotelian tradition. And then there is this entirely non-philosophical, *ordinary* sense of the term, whereby such things as carbon, silver, methane, salt and water count as distinct substances, whereas pebbles, planets, dogs and cats do not.

3　See 'Theories of Matter' [2], in Jeff Pelletier's [3], *Mass Terms, some Philosophical Problems*. The doctrine is represented in Aristotelian terms by the metaphysical primacy of 'primary substances'.

4　For 'how else', he enquires somewhat rhetorically, 'is there to to talk?' [4].

5　A related tendency construes apparently non-singular expressions of the concept as being, in actuality, singular in a semantic sense. This seems to be the case with Frege.

6　Even at the level of referential applications of the corresponding expression, there are differences between the two. Thus the combination of a singular occurrence of a noun with a demonstrative results in the determinate identification of a particular individual, whereas the plural occurrence of a noun with a demonstratve results in no such determinate identification of multiple individuals. In the former case there is unique reference to just one, *exactly one* object, whereas in the latter case, even if a determinate number of items of the appropriate kind are present, the use of a suitable demonstrative—as in 'those birds', maybe, or again with 'these mosquitoes ', the (potential) numerical determinacy of the items indicated is not reflected in any such numerical determinacy of the referring expression. Indeed, while a singular occurrence of a thing-word carries an implication of identity for the corresponding object, a plural occurrence of a thing-word requires *no* implication of identity among the corresponding objects. Whereas

　　　A student has been waiting to see you for the past hour
carries an implication of identity for the designated student,
　　　Students have been waiting to see you for the past hour
carries no such implication of identity—not even for a single student (nor indeed for at least one). Again, to the question
　　　Are any students waiting to see me?
The answer must be 'YES', if there is one or more (i.e., at least one).

7　As the term comes to us from Aristotle, categories are elements of metaphysical or ontological taxonomy—Aristotle's basic category being that of so-called *substance*. Related categories might include those of *physical objects*, *events*, and *attributes*. Although closely related, ontic categories are distinct from logical and semantic categories: *ontic* categories are language-independent; *semantic* categories are plainly not. Ontic categories, while not necessarily the highest taxa of kinds of beings, must at least be higher than any straightforwardly empirical taxonomy of kinds. The question has been recently discussed in some detail by Jan Westerhoff [5].

8　Notice that this mode of designating a concept is not *referential*—for obvious reasons, that is not possible in the case of concepts—but is rather a classical Russellian case of *denoting*.

9　The point is one that leads Quine to engage in 'reducing universals to particulars'; see [7] (p. 98). Thus Quine: 'On the efficacy and limitations of the device as a means of reducing universals to particulars see Goodman, *Structure of Appearance*, pp. 155 f., 203 ff., and my *From a Logical Point of View*, pp. 68–77.' The issue is considered further in the sequel.

10  This work is composed and written in the early days of a time of (potentially terminal) global ecological crisis, centered at this stage on fossil fuels. I do not think philosophy is immune from this crisis; indeed, I believe that the problem this work addresses is *itself* a symptom of this crisis. A possible history might judge as to the origins of the crisis; it might of course be concluded—especially with further relevant evidence—that there are good reasons to believe it is the fate of almost every civilization anywhere. But perhaps most of the others had a special, wise and objectively grounded way of life, *self-consciously* in tune with the overarching ways of the cosmos.

11  And here I take the view that to be extended in time is itself (really, metaphysically) *irreducible*, since time is no more (really) reducible to a set of moments than is space to a set of points. Mathematical points of time are a calculus-type *abstraction* time, rather than its basis. In other words, there is a *genuine* form of self-identity, that of Aristotelian substances, which consists in (absolute) persistence over time. From a temporal standpoint, all concrete predications, whether singular or not, should be considered to have some such rider as 'since a (*finite time stretch*) ago'. (These statements are not ones I shall here try to defend, though they strike me as intuitively plausible).

12  The point is noted, albeit discretely, by Quine, and forthrightly, by Strawson. Quine however, unlike Strawson, proposes to 'reduce' these anomalous universals to particulars. See Quine [7], (p. 91, fn. 3) and Strawson [12], (chp. 6, sec 6, p. 202 ff).

13  Pools and rivers may then be said to *be water*, and rings and lumps be said to *be gold*. But this is not to say that (much like 'red') 'gold' and 'water' are the names of attributes. Instead, these lumps and pools *consist* of gold and water; they have gold and water *in* them.

14  However, what such a formula expresses, in my view, is a relationship of necessary *composition* rather than identity. My realism here is not speculative by intent; it is no more than an attempt to do justice to the category implicit in the language of chemical substances, as grounded in their homogeneous and undifferentiated nature.

15  The distinction between physical and chemical properties of matter is fundamental. The conditions of being solid, liquid or gaseous, along with melting points, boiling points, density and so forth are all physical properties; such phenomena as the oxidation or rusting of iron and the fermentation of barley and grapes are chemical processes, grounded in chemical properties—lawlike powers of interaction with other substances. Unlike chemical properties, many physical properties vary in lawlike ways with external physical conditions (temperature, pressure, and so forth).

16  Milk, butter, glue, wood, hashish and whisky all count as substances in a more informal sense; they are all the correlates of concrete *mass*-words, of which more shortly; and apart from the constituents of organisms (wood, flesh, and suchlike), all are mixtures.

17  With kinds of stuff like hail and snow, it is plausible to regard the corresponding *substance* just as water, such that snow, much like ice, is conceived as a crystalline condition *of that substance.* (At any rate, it does so on this particular planet, as of now. Confronted on an alien world by stuff distinct but phenomenologically indistinguishable from snow on Earth, the most appropriate answer to the question 'What kind of stuff is that?' could well be 'Carbon dioxide'—but had better *not* be 'Snow!'). The issue then is centrally a question of the weight that can be carried by this notion of a kind, when speaking in particular of 'kinds' of stuff. So too with such 'kinds' of stuff as sand and powder, sand being particulate stuff composed of a variety of minerals and mixtures thereof—small particles or grains of different kinds of rock. Nevertheless, classifying sand and whisky as substances seems harmless enough for practical purposes. There is a more informal use of the terminology of 'sorts' and 'kinds', whereby it seems unproblematic to classify sand, whisky, snow and ice as different sorts or kinds of stuff; but the narrow concept of a substance cuts more deeply.

18  Such properties, used in identifying substances, are commonly characterised as *intensive* properties, and contrasted with *extensive* properties such as mass and volume, measures of the amount of substance present, and not relevant to the identity or identification of the substance.

19  The terminology of 'bare' nouns and noun phrases is relatively recent and originates with Chomsky [13]. Sentences of this kind are, or include, what Peter Strawson has called 'feature-placing' sentences. See his *Individual* ([12], chp. 6) and 'Particular and General' ([14], secs. 6 and 7).

20  Notice that since barking is an (essentially extended) *activity*, as against a *state*, times must be understood as *intervals* in which activity occurs, not as dimensionless moments or temporal 'points'. By the same token, truth-values must be assigned to sentences relative to intervals. In fact, it is far from obvious that such things as dimensionless points exist; their mathematical function, in my view, is as ideal objects—purely heuristic devices or abstractions within formal models. On this issue, see especially Prat and Bree's 'The expressive power of the English temporal preposition system', *Technical Report Series* UMC-93-1-7.)

21  I do not, of course, intend to deny that identity enters into the use of the term 'dog', insofar as its *singular* occurrences are necessarily identity-involving or referential; the point concerns plural occurrences alone.

22  In the description of the topic of a workshop on "Indefinite Reference", Barbara Partee writes:One standard view among logicians is that indefinite noun phrases like 'a tall man' are not referring expressions, but quantifier phrases, like 'every man', 'all men', and 'most men'. Yet in many respects, indefinite noun phrases seem to function in ordinary language much like definite noun phrases or proper names, particularly with respect to the use of pronouns in discourse. This may be simply a matter of sorting out semantics from pragmatics, but there is not to our knowledge any currently available theory that simultaneously characterizes the logical or truth-functional properties of indefinite noun phrases and accounts for their 'discourse-reference' properties (Heim [15]). In my view, Partee has a point, and while the non-bare indefinite sentences clearly have existentially quantified truth-conditional

equivalents, they go *beyond* their truth-conditional equivalents in an important way—a way I shall speak of as having *indefinitely referential* or *identity-involving* force. For convenience, I may also speak of the non-bare sentences as having the semantical property of simply *being* referential or identity-involving, but the semantical property in question is probably best understood as I have introduced it—as a semantically *grounded* potential or force, distinct from the actual truth conditions of the sentence.

23　　The claim that I have written *some* books, or *a number of books*, may be misleading although not false, if I have written only two; but the claim is outright *false*, and not merely misleading, if I have written only one. As Strawson notes in *Introduction to Logical Theory*, it is a distinctive feature of 'some' that it 'carries an implication of plurality' ([16], p. 178).

24　　That is, the concept itself—and via the concept, the entire sentence—imposes no quantitative limits on the availability of water, which is not of course to say that it implies a truly limitless supply.

25　　See [11] (p. 198). The use of the term mass in a semantic sense originates with Jespersen in 1913 [18]. The semantic 'mass/thing' contrast is marked syntactically in many human languages, but in numeral-classifier languages such as Mandarin, it is instead contextually indicated.

26　　These expressions have now been replaced by the elegant 'count nouns' and 'mass nouns', although there are, I will suggest, reasons to avoid this particular expression of the dichotomy.

27　　The work in question is, after all, entitled *The Philosophy of Grammar* [11].

28　　It is nevertheless surprising—given both his academic discipline and the unsurprising fact that he seems unaware of the work of Frege, Russell, and Wittgenstein—that Jespersen should speak of the need for an 'ideal language, constructed on purely logical principles'.

29　　In this manner, there seems to be a quasi-metaphysical or quasi-ontic concept underlying Jespersen's category of 'mass-words'; and it is embodied in the idea of material stuff or matter of one kind or another as *unbounded*.

30　　As earlier remarked, bags can be filled not only with sand or flour, but also with clothing or equally with clothes.

31　　At the same time, replacing 'beer' with 'rain', suppose, and 'apples' with 'raindrops' in column three, might suggest a potential real-life intimacy of the non-plural to the plural terms (for the overall range of appropriately cognate expressions). A kind of 'physical' or conceptual 'differentiation' might then be envisaged between the two adjacent cells in the non-singular column.

32　　In 'Function and Concept', as already noted, Frege poses just such an inappropriate question concerning the object-concept—again, the question, he declares, of 'what it is that we are here calling an object' ([19], p. 143).

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
