# Peer review of "The Concept of a Substance and Its Linguistic Embodiment"

_philosophies, doi:10.3390/philosophies8060114_

Round 1
Reviewer 1 Report
Comments and Suggestions for Authors
This paper addresses the issue in philosophical logic and linguistic semantics over count and mass terms. The argument appears to be that the ordinary notions of “stuff” or “substance” are basic to our conceptual scheme but beyond the reach of the standard notions of subject and predicate that underly contemporary logic.
That is a huge topic and is worthy of a book length essay. Indeed, this paper at over 15,000 words, is too long for a standard journal article and may be a first draft of a book. The theses are only hinted at (repeatedly) and only become clear as the excellent and learned prose flows on. Some peculiar diagrams that the author may use when lecturing on this topic may have been explained in a larger manuscript from which this manuscript has been carved. The paper contains minimal references to the literature, but all of it somewhat dated. Quine is the only philosopher dealt with at any length, and the quite references to Vendler, Mourelatos, are indicative of the state of the art some decades ago. Why not discuss Montague semantics rather than Frege when seeking a model of logicians’ accounts of this distinction? (One unexplained reference to Sarah Jane Leslie is the only literature written in this century that I found.)
In one place the paper seems deliberately sly, rather than simply unhelpful and obscure. This is the argument that because physical quantities are studied with real numbers, and Cantor showed that there are uncountably many real numbers, that the logic of counting, of same and different objects due to Frege, does not apply to physical substances like water! The famous Cantor argument, as formulated in set theory or even in a (consistent variant of) Frege’s type theory in fact shows that the real numbers are uncountable in exactly the logic of identity and quantification that the author denigrates. There are two notions of “countable” that the author has confused.
I suppose that a paper of half the length, with a clearly stated thesis, diagrams that are useful or deleted, and with some discussion of recent literature might be publishable in your journal.
Reviewer 2 Report
Comments and Suggestions for Authors
The paper deals with the concept of substance in an ordinary and (as the author thinks) "non-philosophical" sense, in which substance is what is denoted by a mass term (e.g., carbon, silver, water etc.). It is argued that a standard logical-ontological approach stemming from Frege-Russelian tradition cannot provide a sufficent analysis of such a concept of substance. After relatively large introductory historical remarks (pages 1-7) and the presentation of the Jespersen's approach (pages 7-9), the paper aims to describe a specific nature of substance (in the sense of mass terms). Among the key features on which the paper focuses are the lack of both singular and plural form of "unbounded nouns" (denote what is spacio-temporally unbounded, stuff) and the absence of logical subjects in "unbounded noun sentences". The proposed conclusion is that stuff has a "simpler and more fundamental" mode of being than anything else referred to (directly or indirectly, that is, quantificationally) in referential sentences (p. 19). Appendices (pages 22-35) contain additional considerations on the key concepts used (e.g., object, concept, category, number, identity, change).
The paper offers a rich variety of ideas and analyses, which may be inspiring for a deeper research on mass terms and the stuff concept. However, it lacks a clear and strong argumentation line for a clearly stated main thesis, and is rather a composition, sometimes repetitious, of a several thematically interconnected sections and appendices. Some sentences are not quite clear (e.g., footnote 10; text 306-324).
It is not clear why stuff (water etc.) should be substance in a "non-philosophical" sense since the concept of "primordial matter" is clearly an Aristotelian concept influential through centuries. Also, as mentioned in the paper itself ("Historical remarks") , water, air etc. were fundamental concepts for the so-called pre-Socratics. The paper is focused mainly on standard logical-sematical framework, with some reflections, for example, on logic of plural sentences and mereology, but does not take enough into account a variety of other approaches to mass terms (see "The Logic of Mass Expressions" in Stanford Encyclopedia of Philosophy, e.g., Ter Meulen 1981) and objects in general (e.g., objects in modal, many-valued or fuzzy context).
It is not clear what is the scope of "Introduction" subtitle, since the next bold subtitle is "Appendix". The first appendix is not enumerated and the enumeration within the appendices is incorrect (instead of "19.1" there should be "10.1", instead of "9.2" there should be "10.2", Appendix II starts with "2.0"). Footnote 8 is empty. Footnote 13 is repeated in the main text. There is no list of references at the end of the paper, some references in the text are incomplete (e.g., ftn. 1, 13, 16) and some are missing (e.g., ftn. 20 Grice and Dummett; ftn. 22 Brecht; lines 889-890 on logic of plural sentences). There are some typos (e.g., "howver", line 138; "and and" footnote 14; "6" line 63, punctuation missing, ftn. 12, 26, 38).
The impression is that the paper is not re-read carefully enough and is not given an appropriate firm argumentative and presentation structure. Probably, the material of this paper could be carefully reworked, on the ground of some further study, and thoroughly argumentatively restructured into an essentially new paper.
Round 2
Reviewer 1 Report
Comments and Suggestions for Authors
See attached file.

Author Response
The chief concern of Reader 1 related to typos and inadequate references. These have all been addressed. So I do not think there is more to be done in this connection.
Reviewer 2 Report
Comments and Suggestions for Authors
The author's response is not acceptable for me.
Author Response
The referee does not approve of my work, but since they provide no reasons for this view, I cannot take their opinion seriously.